gb4gv: a genome browser for geminivirus

Ho Eric S. hoe@lafayette.edu 1 2
Newsom-Stewart Catherine M. 1
Diarra Lysa 1
McCauley Caroline S. 1
1 Department of Biology, Lafayette College , Easton , PA , United States
2 Department of Computer Science, Lafayette College , Easton , PA , United States
Grande-Pérez Ana
Electronic publication date: 2017 Apr 12
Publication date: 2017
Volume: 5
Electronic Location ID: e3165
Received 2017 Jan 5; Accepted 2017 Mar 9
Copyright: ©2017 Ho et al.
Copyright year: 2017
Copyright holder: Ho et al.
License: This is an open access article distributed under the terms of the Creative Commons Attribution License, which permits unrestricted use, distribution, reproduction and adaptation in any medium and for any purpose provided that it is properly attributed. For attribution, the original author(s), title, publication source (PeerJ) and either DOI or URL of the article must be cited.
License URL: https://creativecommons.org/licenses/by/4.0/

Keywords: Alphasatellite, Betasatellite, Begomovirus, Curtovirus, Geminiviridae, Geminivirus, Mastrevirus, UCSC Genome Browser

Funding: Lafayette College This project was supported by the startup fund provided by Lafayette College to ESH. CMN was supported by the EXCEL Summer Scholar Program funded by Lafayette College. There was no additional external funding received for this study. The funders had no role in study design, data collection and analysis, decision to publish, or preparation of the manuscript.

==============================
Background

Geminiviruses (family Geminiviridae) are prevalent plant viruses that imperil agriculture globally, causing serious damage to the livelihood of farmers, particularly in developing countries. The virus evolves rapidly, attributing to its single-stranded genome propensity, resulting in worldwide circulation of diverse and viable genomes. Genomics is a prominent approach taken by researchers in elucidating the infectious mechanism of the virus. Currently, the NCBI Viral Genome website is a popular repository of viral genomes that conveniently provides researchers a centralized data source of genomic information. However, unlike the genome of living organisms, viral genomes most often maintain peculiar characteristics that fit into no single genome architecture. By imposing a unified annotation scheme on the myriad of viral genomes may downplay their hallmark features. For example, the viron of begomoviruses prevailing in America encapsulates two similar-sized circular DNA components and both are required for systemic infection of plants. However, the bipartite components are kept separately in NCBI as individual genomes with no explicit association in linking them. Thus, our goal is to build a comprehensive Geminivirus genomics database, namely gb4gv, that not only preserves genomic characteristics of the virus, but also supplements biologically relevant annotations that help to interrogate this virus, for example, the targeted host, putative iterons, siRNA targets, etc.

Methods

We have employed manual and automatic methods to curate 508 genomes from four major genera of Geminiviridae, and 161 associated satellites obtained from NCBI RefSeq and PubMed databases.

Results

These data are available for free access without registration from our website. Besides genomic content, our website provides visualization capability inherited from UCSC Genome Browser.

Discussion

With the genomic information readily accessible, we hope that our database will inspire researchers in gaining a better understanding of the incredible degree of diversity of these viruses, and of the complex relationships within and between the different genera in the Geminiviridae.

Availability and Implementation

The database can be found at: http://gb4gv.lafayette.edu.

Introduction

Geminiviruses (family Geminiviridae) have emerged as one of the most prevalent and problematic plant pathogens especially in developing countries (Sattar et al., 2013; Scholthof et al., 2011; Shepherd et al., 2010). In terms of diversity, they have become the largest group of plant viruses to exist today. It posts significant threat both socially and economically as geminiviruses are the most destructive pathogens in subsistence agriculture like beans, cotton, maize, sweet potato, and tomato (Jeske, 2009; Sattar et al., 2013; Scholthof et al., 2011; Shepherd et al., 2010). The economic impact of geminivirus infection can be seen across the globe. Annual economic loss is estimated to be US $1.9–2.7 billion in East and Central Africa. Maize streak virus alone has caused hundreds of millions of loss in food crops per year (Shepherd et al., 2010).

Geminiviruses often infect plants as complexes: a mixture of viral isolates identified as distinct species, as well as DNA satellites. Moreover, they are able to undergo mutation, recombination and reassortment both frequently and rapidly. Together, these factors increase the diversity and capabilities of the family, allowing them to invade new hosts and new environments without complication. In order to prevent geminiviruses from becoming even more of a threat to our growing human population, it is critical that scientists are able to better understand the genomic sequences of these viruses. Geminiviruses rely heavily on their host’s cellular machinery so having a greater knowledge of their genetic makeup will allow scientists to formulate biotechnological means to help plants fight their attackers successfully.

Geminiviruses comprise a family of plant viruses that exist in the form of twinned icosahedral particles holding small, circular, single stranded deoxyribonucleic acid (ssDNA) genomes. The ssDNA genome structure enables it to evolve at high rate comparable to RNA viruses (Duffy, Shackelton & Holmes, 2008). The viral genome encodes only 5–7 proteins, making geminiviruses one of the smallest virus types known to scientists today. Within Geminiviridae, seven genera are currently recognized by The International Committee on Taxonomy of Viruses (ICTV): Becurtovirus, Begomovirus (the one with the largest number of species), Curtovirus, Eragrovirus, Mastrevirus, Topocuvirus, and Turncurtovirus. Depending on the genera, the viral genome comprises of either one (monopartite) or two (bipartite) DNA components. Monopartite genomes consist of a circular, ssDNA molecule (similar to the DNA-A component of bipartite genomes) that is often associated with an alpha- or betasatellite, while bipartite genomes consist of separated DNA-A and DNA-B components of similar size.

National Center for Biotechnology Information (NCBI) designates a separate website to host viral genomes (National Center for Biotechnology Information, 2016d). Its collection includes almost all known viruses in the world, making it one of the most comprehensive resources for studying viral genomics. Viral genomes are formatted in a standard GenBank record (National Center for Biotechnology Information, 2016a) exactly like other living organisms. However, genome architectures of viruses exhibit significant difference from living organisms. For instance, virions of bipartite begomoviruses encapsulate two circular, ssDNA components in which the two components are required for infectivity. But such critical association between the two components is often missing from NCBI Viral Genome database. Moreover, vital information about the virus such as location where it was found, targeted hosts, etc. are not searchable attributes, limiting the utility of the database. These are the reasons that we have undertaken this project in providing researchers a comprehensive, up-to-date, and integrated environment at their fingertips. The database we built rests on the software architecture of UCSC Genome Browser website (Kent et al., 2002; Kent et al., 2002) as such we named our database gb4gv, which stands for Genome Browser for Geminivirus. For clarity, we reserve “UCSC Genome Browser” to refer to the website itself (Kent et al., 2002), and “Genome Browser” to mean the software that supports the website. Genome Browser was chosen because of its versatility in visualizing genomes, richness in built-in functions, flexibility in incorporating annotations, and software robustness in handling large volume of requests—872,000 requests per day on average (University of California Santa Cruz, 2016b). Although Genome Browser offers these benefits, its original design gears mainly toward eukaryotes. In order to unleash the power of Genome Browser, we have made substantial effort in modeling geminivirus genomes into a structure that can take full advantage of its functionalities. gb4gv can be accessed, without registration requirements, from here: http://gb4gv.lafayette.edu. Users can make use of the built-in functions provided through our website to download genomes or sequences of interested regions freely.

Materials & Methods

Compilation of geminivirus genomes

Figure 1 summarizes the semi-automatic annotation workflow designed for this project. The primary source of our data originated from NCBI RefSeq database (National Center for Biotechnology Information, 2016c) because redundant genomes were purged. But we also cross-referenced our data with the ICTV Master Species List 2015 v1 obtained from ICTV (2015). We had identified 700 RefSeq entries that comprised of 529 distinct geminiviruses. Note that DNA-A and DNA-B of a bipartite begomovirus are kept in separate entries in RefSeq. Begomovirus occupies the largest genus of the family, followed by Mastrevirus, and Curtovirus. Other genera were found sporadically including two becurtoviruses, one topocuvirus, one eragrovirus, and one turncurtovirus, while genera of ten entries remain unknown. Here we had decided to incorporate only genera that represent major Geminiviridae genera i.e., Begomovirus, Curtovirus, and Mastrevirus into gb4gv. As a result, genomes of 514 geminiviruses representing 97% of the Geminiviridae found in NCBI were considered for further review. We will regularly assess the need to include other minor genera into our database if more samples from them are discovered in the future.

Figure 1 Project workflow of the semi-automatic annotation process.

Our workflow begins with a manual process to identify information generally not documented in the GenBank record such as the acronym of the virus, location, infecting host, monopartite or bipartite genome, and genomes association for bipartite virus. This information is passed to a downstream automatic process that integrates them with other sources. The automatic procedure parses GenBank entries from RefSeq database for genomic information of geminiviruses including the accession numbers, genomic sequences, genes, viral proteins, and taxonomy ID. siRNAs from host plants that fight against viral infection were obtained from NCBI GEO database. In the last step, geminivirus information is formatted into UCSC Genome Browser format.

Besides the main genomes, ancillary alphasatellites and betasatellites are often isolated together with monopartite begomoviruses (Xie et al., 2010) and they are found to play essential roles in boosting host’s symptoms and viral movement (Briddon et al., 2001; Saunders et al., 2004; Zhou et al., 2003). We had identified and reviewed 66 and 105 alphasatellite and betasatellite genomes, respectively, from NCBI.

Meta information or attributes such as the geographical location of the virus are important to understand the virulence of the virus but it is not always available in genome database. Therefore, we manually searched for additional information about these viruses from existing literature. In particular, we focused on identifying or reconfirming the location where they were collected, the hosts they infected, their acronyms, monopartite or bipartite genome, and the counterpart genome in case of bipartite. Importantly, we have made these attributes searchable in our database.

Following the manual process is the automatic annotation process. In this step, NCBI RefSeq entries belonging to Geminivirus were parsed to ensure that each entry satisfies the following two criteria:

1. Every geminivirus and geminivirus-associated DNA satellite genomes must possess the iconic structurally conserved element (SCE), which is the genomic landmark of geminiviruses including satellites. The canonical structure of the SCE is TAATATT—AC, where “—” stands for the cleavage site targeted by the viral replication protein in the initial step of DNA replication (Gutierrez, 1999; Jeske, Lutgemeier & Preiss, 2001; Pilartz & Jeske, 2003). The prevalent SCE sequence of alphasatellite is TAGTATT—AC, which varies slightly from the canonical SCE sequence. Nonetheless, owing to either DNA sequencing errors or random mutations, the 5′ side of the SCE of some viruses may deviate slightly (less than one nucleotide) from the canonical form from above. To accommodate such minutiae, we tolerated entries with up to one mismatch from TAATATT. Genomes failed to meet this criterion were excluded from gb4gv.

2. Besides genomes, gb4gv also keeps individual viral proteins if they satisfy our quality checking. The coding region (CDS) of a gene defined in a RefSeq entry must be translated exactly into the stated peptide in the RefSeq entry. Genes failed this criterion were excluded from our database. But genomes containing erroneous CDS were still kept in the database.

Through our tandem manual and automatic annotations, six out of 514 RefSeq entries of geminiviruses failed the validation process stated above, resulting in 508 genomes being selected into our database. For satellite genomes, seven out of 66 alphasatellites and three out of 105 betasatellites failed our validation. Table 1 categorizes all the genomes accepted into our database by genus, number of genomes per virus, and geographical origin. The aforementioned annotation information can be downloaded from our website in tab-separated format (http://gb4gv.lafayette.edu/downloads.html).

Table 1 A summary of genomes stored in gb4gv.

The numbers inside the parentheses denote the numbers of genomes. The lower part of the table categorizes begomoviruses further by origin and the number of genomes per virus.

Geminiviridae (508)	Satellite (161)	
		Begomovirus (470)			
Curtovirus	Mastrevirus	DNA-A	DNA-B	Alphasatellite	Betasatellite	
5	34	338	132	59	102	
Begomovirus (338)	
Old World (216)	New World (119)	Unknown origin (2)	
Monopartite	Bipartite	Unknown	Monopartite	Bipartite	Unknown	Monopartite	Bipartite	
100	44	72	12	95	13	1	1	

Small Interfering RNAs

A key aspect of gb4gv is to inspire researchers to formulate insightful strategies that can be used to eradicate the propagation of geminiviruses. Therefore, studying the immune response launched by infected plant is a promising research direction. Thus, we had downloaded datasets from two small interfering RNAs studies deposited in the NCBI GEO database (National Center for Biotechnology Information, 2016b): GSM425427, and GSE26368. siRNA sequences were mapped to the genomes of begomovirus and betasatellite through a customized Python script. Mapping tolerated up to two mismatches in internal positions without gaps. A Genome Browser annotation track is designated for each sample, which can be found under “Mapping and Sequencing” section of each virus. In Begomovirus or betasatellite, six siRNA tracks were configured.

Standardization of circular genomes

Like many other genomic databases such as NCBI RefSeq and UCSC Genome Browser, circular genomes are linearized. Instead of opening the circular genome at arbitrary sites, circular genomes were opened at the biological cleavage site at the SCE. A benefit of standardizing the opening site is to facilitate syntenic analysis (to be discussed in Multiz section). Under the standardized linearization scheme, a genome always begins with ‘AC’ and terminates with ‘TAATATT’ at the 5′ and 3′ termini, respectively. Thereby we standardized all genomes obtained from RefSeq. Genomes not conforming to this standard were shifted until they met the above criterion. Out of 669 accepted genomes in gb4gv, surprisingly, 112 (17%) of them required this adjustment.

Data Models

Genome Browser was originally designed to visualize mammalian genomes (Kent et al., 2002). It was later enhanced to host non-mammalian animals e.g., C. elegans, and then unicellular organisms such as yeast. Ebola genome is the first and remains to be the only viral genome available in UCSC Genome Browser at present. This historical background reveals that the data model of Genome Browser is geared toward the display of chromosomes of a species. Such data model serves well with living organisms but it poses two challenges in configuring Genome Browser for geminivirus genomes:

1. The genus Begomovirus is known to be diverse (Brown et al., 2015) with over 300 DNA components being identified by us. If we were to coerce the existing data model to begomoviruses, 300 databases are needed, leading to a huge species tree in the home page, hampering website performance, and prohibiting data browsing. To circumvent this, we modeled each viral genus as an organismal species, and the array of viral species of a genus as chromosomes of an organism. Based on this workaround, gb4gv consists of five databases (a database per genus including one for each satellite although, in biological terms, satellite is not considered a genus): Begomovirus, Mastrevirus, Curtovirus, alphasatellite, and betasatellite.

2. A special configuration is needed to establish the association between the bipartite DNA components (DNA-A and DNA-B) of a begomovirus. In gb4gv, DNA-A and DNA-B were treated as two separate chromosomes. The coupling of DNA-A and DNA-B components of a bipartite begomovirus can only be achieved manually as their RefSeq accession numbers reflect no information about their relationship. In order to facilitate users to associate them easily, a viral species in our database is uniquely referenced by an acronym, e.g., AbMBV is the reference of Abutilon mosaic Brazil virus. But the two DNA components of a bipartite begomovirus will become indistinguishable under this scheme. Thus, we suffix the acronym of a bipartite virus by “.A” and “.B”. For example, the DNA-A and DNA-B of the virus AbMBV can be found effortlessly through AbMBV.A and AbMBV.B, respectively. An advantage of using an acronym as the key to retrieve a virus is to release the burden of users to pull up the accession number of the virus as most people can remember the acronym rather than the arbitrary accession number. Moreover, we recognize that some viruses are referenced by multiple acronyms without a consensus. To accommodate such variability our database maintains a list of searchable aliases for every virus; for example, Tomato leaf curl New Delhi virus can be identified by either TolCNDV or ToLCV_India.

Common region identification in bipartite begomoviruses

The bipartite genomes of a begomovirus share a highly similar, non-coding segment flanking the SCE “TAATATTAC”. This segment is colloquially named the common region (CR). CRs serve a crucial role in viral DNA replication. Studies had shown that the 5′ side of CRs contain replication protein binding sites (Orozco & Hanley-Bowdoin, 1996). Thus, CRs harbor vital regulatory signals that influence the replication and the coupling of the bipartite genomes for begomoviruses. Understanding viral replication is fundamental to combat viral infection. Thus, we undertook the task to predict CRs in bipartite begomoviruses. Based on the manual annotation we did, DNA-A and DNA-B components of a begomovirus were paired up. We extracted the non-coding region, also known as the long intergenic region (LIR), between REP and CP genes in the DNA-A or between NSP and MP genes in the DNA-B. In the next step, we further reduced the LIR into an 809-bp segment, which consisted of a 400-bp segment upstream and downstream of the SCE from the DNA-A and DNA-B. In Fig. 2, two 809-bp segments were aligned by MUSCLE (Edgar, 2004) as shown:

Figure 2 Identification of common region (CR) shared between DNA-A and DNA-B of bipartite begomoviruses.

Sequence alignment of two 809-bp segments located in the LIR of the Old World bipartite East African cassava mosaic Kenya virus (EACMKV). The invariant SCEs are highlighted in red. The inverted repeats constituted the stem of the hairpin structure are highlighted in blue. The two underlined regions indicate the 5′ and 3′ termini of the common region determined by our method of using a 20-bp sliding window.

In this example, the two segments were extracted from DNA-A (NC_011583) and DNA-B (NC_011584) of the Old World bipartite East African cassava mosaic Kenya virus (EACMKV) and they were aligned. A 20-bp sliding window was used to scan the alignment base-by-base bilaterally starting from the SCE. Scanning halts when the percentage of sequence identity within the window drops below 80%, an adjustable parameter. The halting locations (the underlined regions in Fig. 2) are taken as the 5′ and 3′ termini of the common region.

The average size of a CR was found to be 212 bps (including the SCE) in which the 5′ arm, the left segment of the SCE, is usually longer than the 3′ arm with an average size of 150 bps. The longest CR is 417–419 bps long that belongs to Indian cassava mosaic virus (NC_001932/NC_001933). Whereas Abutilon mosaic Brazil virus (NC_016574/NC_016577) was found to possess the shortest CR, which is 63–67 bps long. Also note that the two approximately 10 bps segments juxtaposing the SCE constitute the stem part of the hairpin structure (Fig. 2).

Putative iterons and TATA box sites

One of the cis-regulatory signals harboring in CR is iterative elements, also known as iterons (Arguello-Astorga et al., 1994; Arguello-Astorga & Ruiz-Medrano, 2001; Sanz-Burgos & Gutierrez, 1998). A distinct feature of iterons is the presence of direct or inverted sequence repeats. The following rules were applied to predict iterons in viral genomes:

1. They are located in the 5′ side of the LIR i.e., from the beginning of the first gene on the complementary strand up to, but excluding, the SCE

2. The minimum length of an iteron is 9 bps. There is no restriction on the maximum length

3. The pair of repeats identified differ by at most one base

4. Repeats could be direct or inverted

5. Its location is no more than 100 bps from a putative TATA box, if any

6. At least one of the twelve iteron core motifs is present (Arguello-Astorga & Ruiz-Medrano, 2001): GGAGN, GGTAV, GGGGW, GGTAV, GGKGT, GGKGG, GGGGG, GGGGA, GGGTM, GGCGT, GGWGT, and TGGTGTCC.

As the TATA box sites and iterons work cooperatively to regulate replication, we also identified putative TATA box sites. The consensus sequence of a TATA box is defined as “TATA”, followed by any number of “TA” or “AA” repeat (Bernard, Brunaud & Lecharny, 2010; Patikoglou et al., 1999). We developed a Python script to scan for iterons and TATA box sites in every geminivirus genome. Based on the above criteria, 5,142 iterons and TATA box sites were predicted from 669 components and genomes. Results can be visualized in gb4gv by activating the ‘Iterons and TATA” annotation track.

Multiz track

Genomes of various species within a genus share similarities and differences. Since we have standardized the opening site of the viral circular genomes, a genus-wide syntenic analysis becomes possible. Such comparative view helps to uncover conserved and diverse genomic regions among species. We used the threaded blockset aligner (TBA) (Blanchette et al., 2004) to generate a dynamic multiple sequence alignments of all species in a genus. Unlike other multiple sequence alignment programs of which a sequence from the sample is dedicated to be the reference of the alignment, TBA produces a multiple sequence alignment dynamically based upon the genome being selected for viewing in the Genome Browser. This unique feature enables gb4gv to generate a graphical representation regarding genome conservation among different species with respect to the current queried genome.

TBA requires two mandatory inputs: a set of genomic sequences, and a phylogenetic tree defining the evolutionary relationship of the input genomes. We used multiple sequence alignment program MUSCLE (Edgar, 2004) to build phylogenetic trees, followed by maximum likelihood tree building PHYLM (Felsenstein, 2005) equipped in MEGA7 (Kumar, Stecher & Tamura, 2016). The output phylogenetic tree was in NEWICK format. Based on the genomic sequences and the phylogenetic tree, TBA generated the threaded blockset alignment. The alignment was loaded to a MySQL database referenced by Genome Browser.

UniProt/SwissProt annotations

Protein domain information was overlaid on viral proteins in gb4gv. Reviewed Swiss-Prot annotations were downloaded from UniProt website in XML format (UniProtKB). Viral taxonomy IDs served as the key to retrieve protein domain information from the Swiss-Prot annotations. Sequence of the protein domains identified in the search process was mapped to the genomes by BLAT (Kent, 2002).

Genome browser

Version 334 of the Genome Browser was used to build gb4gv. The software was downloaded from the UCSC Genome Browser website (University of California Santa Cruz, 2016a) and installed in our 24-core Linux server running on Centos OS 6.8, Apache 2.2, and MySQL server 5.5.50.

Results

The web interface of gb4gv is organized in a hierarchy consisting of three levels. The highest level presents all the genera of Geminiviridae maintained in gb4gv including two satellites despite they are considered as genera (Fig. 3A). Figure 3B displays information about the genome of an individual virus and corresponding annotation tracks. Detailed information about a particular annotation e.g., a gene, a protein or a specific genomic sequence, is presented at the lowest level (Fig. 3C)

Figure 3 Web interface of gb4gv.

(A) The home page of gb4gv. The evolutionary tree under represented species shows the available viruses including the three genera of Geminiviruses and two satellites. Users can view a particular genus or satellite by clicking on the virus or satellite name in the evolutionary tree. Users can also make use of the “Species Search” box to look up for a particular virus by keywords. Additionally, users can enter keywords in the “Position/Search Term” box search for a particular virus and/or gene. Click the blue GO button to navigate into the genomic information of a particular viral species. (B) This page provides various annotation information about the selected genome in which they are organized in tracks. (C) Information of a protein-coding gene. It tells the genomic location of the gene, size, and strand that codes for the protein. In addition, there is a short description about the current gene including the name of the protein, whether the virus is a New World or an Old World virus, the full name of the virus and its acronym, the RefSeq and GenBank accession numbers with hyperlink linked to the corresponding GenBank entry in the NCBI website.

In the following subsections, we will highlight the unique features offered by gb4gv that are helpful in studying the genomics of geminiviruses. While the software architecture of gb4gv is based on Genome Browser, the operations of our website is highly similar to UCSC Genome Browser. Therefore we will not discuss the data models and functionalities of Genome Browser in details. For readers who are interested in learning more about Genome Browser, we recommend that they consult the online User Guide (University of California Santa Cruz, 2016c).

Search by acronym, accession number, and attributes

To our best knowledge, there is no database that allows users to search for geminivirus genomes or proteins by acronym, host name, geographical location, monopartite, bipartite, Old world, New world, or combinations thereof. For instance, a search for monopartite begomoviruses that infect Okra by the query “monopartite okra” against NCBI RefSeq database returned only two entries: NC_005954 and NC_005051 and both of them belong to satellite genomes. In fact, four monopartite begomoviruses are known to infect Okra according to gb4gv: OLCCV (NC_014745), OYCrV (NC_008377), OYVMV (NC_004673), and OkLCuV (NC_013017). The main reason is because NCBI’s query matches only words in the description of GenBank entries. Our augmented search capability will help researchers in identifying a regime of viruses that share certain attributes handily. gb4gv achieves this by making the above viral attributes searchable in our database in conjunction with the keyword searching capability provided by Genome Browser. Table 2 summarizes the searchable attributes supported by gb4gv.

Table 2 Searchable attributes in gb4gv.

Attribute	Description	Example	
World	Geminiviruses are commonly categorized into “Old World” and “New World” according to the geographical location they were found. This attribute must be either “Old World” or “New World”	old world	
Number of main genomes	It must be either monopartite or bipartite	bipartite	
Acronym of the virus	For begomoviruses, it could be suffixed optionally by “.A” or “.B” to indicate DNA-A or DNA-B of the bipartite genome, respectively.	OMoV.A	
Host	Name of the host infected by the virus	Okra	
Country	The country that the virus was found	Brazil	
RefSeq accession number	The accession number assigned by NCBI RefSeq database	NC_011181	
GenBank accession number	The accession number of the GenBank record that RefSeq used	EU914817	

For instance, to find all begomoviruses that infect sweet potato, user can input the phrase “sweet potato” in the query box and click the “go” button (Fig. 4A).

Figure 4 Keyword search results.

(A) Search by host e.g., “sweet potato”. (B) Search by a phrase e.g., “new world monopartite”.

User can combine multiple search attributes in a query. The logical AND relationship is assumed between attributes. For example, user can enter “new world monopartite” in the search box to search for all New World monopartite begomoviruses (Fig. 4B). But the current version of the search function remains primitive as it is virtually inherited from the ‘LIKE’ search of MySQL, meaning that the order of queried attributes is important. When multiple attributes are specified, they must be arranged according to the order enlisted in Table 2 from top to bottom. For the same example above, the query “monopartite new world” will result in no hits.

Short match

The ability to support ad-hoc sequence search can help researchers to identify potential short regulatory sequences that can be validated further by experiment. Examples of these regulatory sequences include TATA box (Sanz-Burgos & Gutierrez, 1998), and polyadenylation signal AWTAAA (W means A or T). The Short Match function allows users to search for DNA sequences from two to 30 bases with the support of IUPAC ambiguity codes. Figure 5 illustrates how to specify a short sequence match, and how to inspect the context of a hit within a specific region through the Genome Browser’s zoom-in function.

Figure 5 Setup of Short Match function.

(A) Turn on the “Short Match” track to “full”, and click the Short Match link. It allows users to input the sequence to search for. (B) After clicked the submit button, Genome Browser will return to the main genome view. If the searched sequence is found, results are displayed under the “Short Match” track including the genomic locations prefixed by a + or –to indicate the hit lies in the reference strand or the complementary strand, respectively. (C) Users can zoom in to a smaller region by dragging the mouse pointer.

Putative iterons and TATA

It has been known iterons contributed to viral replication (Arguello-Astorga et al., 1994; Sanz-Burgos & Gutierrez, 1998). Studied had shown binding activities between REP and iterons in Mastrevirus and begomovirus (Fontes, Luckow & Hanley-Bowdoin, 1992; Sanz-Burgos & Gutierrez, 1998). gb4gv maintains 5,142 putative iterons and TATA box sites in the long intergenic region. Users can view this information by turning on the “Iterons and TATA” track. Figure 6 shows an example of iterons and TATA box sites predicted in begomovirus Melochia yellow mosaic virus (NC_028143).

Figure 6 Iterons and TATA track.

Different colors are used to denote various sequence features: direct repeats in blue, inverted repeats in blue–green, and TATA box in red. Tandem repeats are highlighted with “..” at the end the label e.g., the direct tandem repeats “GAATTGGAGTA..” above consists of “GAATTGGAGTATTGGAGTA” in which “GAATTGGAGTA” overlaps with “GaATTGGAGTA” with their overlapping regions underlined. Lastly, our database also highlights palindromic-like sequence by “>>>...>>>”, e.g., “GGAGACTCC”.

Small interfering RNAs

Understanding plant immunity is the foremost step to fight against viral infection. Virus-derived RNA silencing is a vital immune response triggered in plants in the face of viral infection. Thus we have incorporated datasets from two virus-derived small interfering RNA (siRNA) studies into gb4gv. One study used pyrosequencing to sequence siRNAs in tomato leaves (Solanum lycopersicum) inoculated with monopartite begomovirus TYLCV (Donaire et al., 2009). Another study had used deep sequencing to survey siRNAs in the leaves of tomato (Solanum lycopersicum) and tobacco (Nicotiana benthamiana) inoculated with monopartite begomovirus and its associated betasatellite (TYLCCNV/TYLCCNB) (Yang et al., 2011). Both studies had mapped the siRNAs to the genomes of respective hosts. However, it is unclear whether or not these siRNA sequences are species specific. Are siRNAs mapped to biased locations? In order to answer these questions, we incorporated siRNA sequences from these two studies into gb4gv and mapped the siRNAs to genomes of begomovirus and betasatellites. Each sample occupies a track (Fig. 7A).

Figure 7 siRNA mapping.

(A) Six samples of siRNA sequences are available for visualization, one track for each sample. (B) An example to visualize the mapping of siRNAs from GSM425427 on monopartite begomovirus TYLCV. ‘Squish’ mode was used in this example. (C) Another example to show the appearance when ‘Dense’ mode was used to display siRNAs mapped to betasatellite TYLCVVSDB based on samples from GSE26368.

siRNAs mapped to the viral strand and complementary strand are encoded in dark and light color, respectively (Fig. 7B). According to our limited browsing, siRNAs do not map uniformly along the genome. In betasatellites, a sizeable number of mapped siRNAs were skewed toward a 100-bp region near to the 5′ side of the SCE.

BLAT

Our database is also equipped with a lightweight sequence query engine BLAT (Kent, 2002). BLAT stands for BLAST-like alignment tool. It has been widely used to search for highly similar gapped alignments. In situation like the detection of exons based on a spliced mRNA sequence, BLAT provides a speedy mapping of the query sequence onto the genome. Major differences between BLAT and Short Match are:

1. The minimum and maximum query lengths for BLAT are 20 and 25,000 bps, respectively.

2. BLAT searches against genomes in a database specified by the user. Whereas Short Match searches for queried sequence only in the current active genome.

3. BLAT can handle gapped hit but not for Short Match.

As an illustration, we used an unusually long (46 bps) iteron sequence “TGAGTGATTTCTTATTATGTGATTGTCCATTAAAGGGATAAAGTGA” (Fig. 8A) found in YOM (Cotton leaf curl virus betasatellite NC_017829) to query against betasatellite genomes. Intriguingly, eight other betasatellite genomes were found to contain sequences that share from 88.5% to 97.9% of identity with the queried sequence (Fig. 8B). To further examine the hit in virus LPALDDBV, we clicked the “browser” link, which led to Fig. 8C. It shows that the queried sequence hits a region LPALDDBV clustered with iterons. The solid grey bar at the bottom indicates that YOM and LPALDDBV differ at only eight sites.

Figure 8 BLAT search.

(A) Sequence search box. (B) List of hits with hyperlinks referenced the genome browser “browser” and “details”. (C). Display of hit in the context of targeted genome by clicking the “browser” link. The grey bar at the bottom shows the queried sequence is aligned to the opposite strand of the genome. (If the queried sequence aligns to the reference strand, the bar will be displayed in black). Nucleotides displayed on the grey bar represent mismatched nucleotides between the queried sequence and the targeted genome.

Conclusion

Genomics visualization is a useful approach to enhance interpretation especially when the quantity or diversity of the viral genomic data is large. We have harnessed the capability of the widely acclaimed Genome Browser specially for the geminivirus research community. Instead of using a generic one size fits all approach to organize viral genomes, we have taken a semi-automatic pipeline to preserve the unique characteristics of geminiviruses in our web-based database gb4gv. Additionally, we have augmented keyword search capability of manually curated attributes such as infecting hosts, geographical location etc. However, further improvement is needed to accommodate even more flexible multiple attributes queries. Moreover, we have predicted 127 pairs of common regions pertaining to bipartite begomoviruses. This is a useful piece of information as the common regions are implicated in coupling the two main genomes of a bipartite begomovirus during encapsidation. As the ultimate goal in studying the genomes of Geminiviridae family is to understand the underlying genomic features that are suggested to promote its propagation, we have developed our own method to unravel putative iterons and TATA box sites in the 5′ side of the common region and they can be visualized readily with genomic features flanking them. The iterons we predicted are longer (9–46 bps) than the iteron core motifs, GGN1N2N3, reported by this group (Arguello-Astorga & Ruiz-Medrano, 2001). The presence of sizable direct or inverted repeats in fast evolving ssDNA viruses like geminiviruses is unusual, suggesting the existence of negative selective pressure although the biological function of the region peripheral of the iteron core motifs remains largely unknown.

Geminiviruses are diverse and fast evolving. Facilitated by the ever-decreasing DNA sequencing cost, we anticipate more viral genomes will be sequenced in the near future. We are certainly committed to maintaining the information in gb4gv as up-to-date as possible. Given the flexibility of the Genome Browser in accommodating new annotation tracks, if more genome-wide experimental data is available in the future such as Chip-Seq, it can be included into gb4gv readily without software modification as illustrated by the siRNA tracks discussed above. While viral regulatory elements play crucial roles in influencing replication and transcription in cellular environment, we will continue our effort in developing new methods to identify essential sequence elements that might offer new insights for experimental virologists to design effective modalities to fight against the infection of geminiviruses.

We thank the UCSC Genome Browser team for their technical support, especially Maximilian Haeussler, Matthew Speir, and Cath Tyner. gb4gv would not be possible without their kind support.

Additional Information and Declarations

Competing Interests

Author Contributions

Data Availability

The authors declare there are no competing interests.

Eric S. Ho conceived and designed the experiments, performed the experiments, analyzed the data, contributed reagents/materials/analysis tools, wrote the paper, prepared figures and/or tables, reviewed drafts of the paper, build and maintain the web service.

Catherine M. Newsom-Stewart performed the experiments, analyzed the data, wrote the paper, prepared figures and/or tables, reviewed drafts of the paper.

Lysa Diarra and Caroline S. McCauley performed the experiments, analyzed the data, reviewed drafts of the paper.

The following information was supplied regarding data availability:

All data can be downloaded freely from our database website: http://gb4gv.lafayette.edu/downloads.html.

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
