# Peer review of "gb4gv: a genome browser for geminivirus"

_PeerJ, doi:10.7717/peerj.3165_

## Round 0.1 · original submission · Minor Revisions

Thank you very much for your manuscript. Two referees have agreed GB4GV is a valuable tool that will be of help to the research community. However, they raised a number of issues that need to be carefully addressed. I am sure the manuscript will benefit from their constructive advices. I will be happy to receive an improved version of the manuscript, with a point by point answer to their suggestions and comments, before taking a final decision.

Reviewer 1 ·

Basic reporting

The authors present a novel database for geminiviruses, based on the software architecture of UCSC Genome Browser, named GB4GV, which allows the study of the family Geminiviridae with new and useful resources to search and compare viral genomes. The authors have successfully implemented a search tool for valuable information such as hosts and geographical distribution, the existence of putative iterons and TATA-box sequences, identification of common regions; and an important siRNA database obtained from geminivirus infected plants.
The manuscript is clearly written in a professional, unambiguous language and the structure conforms to PeerJ standards.
The specific bibliography of geminivirus in the introduction should be improved by including more references, especially between lines 52-59. On the contrary, Varma & Malathi 2003 does not seem to be a good one as contains obsolete information. Some interesting references to include in the manuscript could be Mansoor, S. et al. Geminivirus disease complexes: the threat is spreading. Trends Plant Sci. 11, 209–212 (2006); Navas-Castillo, J. et al. Maize streak virus: an old and complex ‘emerging’ pathogen. Mol. Plant Pathol. 11, 1–12 (2010); Sattar, M. N. et al. Cotton leaf curl disease — an emerging threat to cotton production worldwide. J. Gen. Virol. 94, 695–710 (2013); and Scholthof, K. B. et al. Top 10 plant viruses in molecular plant pathology. Mol. Plant Pathol. 12, 938–954 (2011).

This DB4GV database, as the authors conclude, could become an important tool for future basic geminiviruses studies.

Experimental design

This manuscript presents numerous strengths:

1. Materials and methods are described with enough detail and following the development of the database is easy.
2. Standardization of geminivirus circular genomes as well as of the first nucleotide of all sequences permit a more realistic comparison of viral sequences.
3. The adaptation of Genome Browser to viruses, modeling each viral genus as an organism species, and the array of viral species of a genus as chromosomes of an organism, is an imaginative solution to facilitate the creation of this database.
4. The association between A and B genomes in bipartite viruses and the possibility of searching for geminiviruses by hosts and geographic location gives a great versatility to this geminivirus database.
5. Both the tool and its interface are friendly and intuitive, allowing an easy search experience.

However, it presents certain weaknesses that should be solved to improve it:

1- In order to become a reference in the field, DB4GV should not exclude more than half of the genera that make up this family. For future updates a priority would be to include databases of the genera Becurtovirus, Eragrovirus, Topocuvirus and Turncurtovirus.
2- Due to the conversion of genus-species and genus-chromosomes, some of the tables generated automatically by the database, i.e. "Schema for siRNA", shows incorrect field titles like "Chromosome" instead of "Genome". Moreover, Genome Browser’s default Help, has this kind of errors too. In the case the authors would not be able to amend it, to avoid misunderstandings this reviewer thinks that it would be interesting that the database interface warned about it.

Validity of the findings

The search by host and by geographical area does not seem complete. For example, a quick search for viruses infecting Solanum sp. retrieves a list in which TolCNDV (Tomato leaf curl New Delhi virus) is left out. And another similar search with "Spain" as geographic location does not include TYLCAxV (Tomato yellow leaf curl Axarquia virus) and others. This point should be corrected or else indicated in the manuscript that this kind of search could give incomplete results.

In the last years we have witnessed a huge increase in of geminiviral sequences uploaded to GenBank due to the rapid geographical expansion and emergence of viruses belonging to the family Geminiviridae. The development of this geminivirus database will with no doubt facilitate their study. The versatility of DB4GV will make it a benchmark in the field. But as the authors remark in their conclusions, the creation of a database not only implies the initial effort creating it but also an explicit commitment to keep it up to date. Therefore, maintenance and improvement of this great tool should be a priority.

Additional comments

N/A

Reviewer 2 ·

Basic reporting

The manuscript by Ho et al. reports the availability of a genomics database for the family Geminiviridae, a large family of single-stranded DNA viruses that infect plants. The database is quite comprehensive and contains useful features that should assist the geminivirus community (and the plant science community in general) in analyzing geminivirus genomics. It is less obvious how the database will help to elucidate the infectious mechanisms of geminiviruses, as claimed (L12-13). Nevertheless, the database is a valuable resource and the manuscript is relevant to inform the community of its existence and features.

The English of the manuscript will require considerable revision (the use of articles is a particularly recurrent problem).

One major issue is a wider acknowledgment of the role of the ICTV as the keeper of official virus taxonomy. Although the NCBI provides an invaluable service with RefSeq, it is the ICTV that recognizes families, genera and species. One problem is that the ICTV does not make a very good job of advertising its activities. However, this may change now that the 10th Report is available as an on line, open access resource (see http://ictv.global/report). The authors should note that currently the 10th Report includes only two chapters, but the <i>Geminiviridae<i/> chapter should be on line any day now, and will include an updated list of all species in the <i>Geminiviridae<i/>, with links to GenBank sequences (and RefSeqs when these are correct).

Another serious problem is the incorrect use of terminology. In fact, I'd say that this is my major issue with the manuscript. I realize that the authors are not virologists themselves, and as someone who has worked on geminiviruses since the 1990's (and have attended every Intl. Geminivirus Symposium since 1995), I can assure the authors that most geminivirologists (like most plant virologists) are quite watchful regarding the use of correct virological and taxonomical terminology. Thus, I urge the authors to correct this problem so that the manuscript is more widely embraced by its main target audience (as stated in L468).

One problem is the confusion between the use of scientific and vernacular names. Scientific names for family and genera must always be italicized and written with a first capital letter: <i>Geminiviridae<i/>, <i>Begomovirus<i/>, <i>Mastrevirus<i/>. There is no plural for scientific names. However, these terms can also be used in the vernacular, and in these situations the words are not italicized and the first letter is not capitalized (unless it is the first word of a sentence, in which case the basic rules of English grammar apply). Also, vernacular words can be singular or plural. Satellites are not officially recognized by the ICTV and thus are not formally classified into families/genera/species. Thus, the words "alphasatellite(s)" and "betasatellite(s)" should always be used in vernacular form, and the names of the satellites should never be written in italics.

Examples:
L1: Geminiviruses (family <i>Geminiviridae<i/>) are prevalent plant viruses that imperil... (note that the words are written correctly in L41)
L18: For example, the virion of begomoviruses prevailing in the Americas encapsulates...
L26: ...four major genera of <i>Geminiviridae<i/>, and...
L46: The economic impact of geminivirus infection...
L56-57: In order to prevent geminiviruses from becoming...
L67-68: Within <i>Geminiviridae<i/>, seven genera are currently recognized by the ICTV: <i>Becurtovirus<i/>, <i>Begomovirus<i/> (the one with the largest number of species), <i>Curtovirus, <i>Eragrovirus<i/>, <i>Mastrevirus<i/>, <i>Topocuvirus<i/> and <i>Turncurtovirus<i/>. (list genera in alphabetical order)
L46, 50, 66, 94, 108, 110, 117, 124, 143, 173, 201, 279, 347, 468: geminivirus(es) (no capital "g")
L118, 128, 167, 214, 226, 233, 234, 242, 415, 418: begomovirus(es) (no capital "b")
L119, 120, 209, 210: genus names in italics
L120-121: ...including two becurtoviurses, one topocuvirus, one eragrovirus and one turncurtovirus, while...
L123: ...<i>Geminiviridae<i/> genera, i.e. <i>Begomovirus<i/>, <i>Curtovirus<i/> and <i>Mastrevirus<i/>, into...
L140: ...belonging to a geminivirus...
L142: Every geminivirus and geminivirus-associated DNA satellite genomes...
L160: ...entries of geminiviruses...
Table 1: family and genus names in italics; replace "Unknown World" with "Unknown origin"
L171, 175, 410: Small Interfering RNAs
L177: ...genomes of begomoviruses and betasatellites...
L180, 202: <i>Begomovirus<i/>
L202-203: That can't be. Maybe the authors identified 500 DNA components (A+B), but not 500 species. The ICTV currently recognizes 322 species (2015 taxonomy). The updated number is probably in the neighborhood of 360.
L231: ...for begomoviruses.
L339: ..genomics of geminiviruses.
L370: ...monopartite begomoviruses (Figure 4B).
L423: ...of begomoviruses and betasatellites.


Additional corrections and suggestions:
L19: ...and both are required for systemic infection of plants.
L32-33: This claim is widely overstrained. My suggestions: ...in gaining a better understanding of the incredible degree of diversity of these viruses, and of the complex relationships within and between the different genera in the Geminiviridae.
L50-52: Delete.
L52: Geminiviruses often infect plants as complexes: a mixture of viral isolates classified as distinct species, as well as DNA satellites. (viruses, not viral species, infect plants; viruses are concrete infectious agents, viral species are man-made abstractions; you eat an orange, not a Citrus sinensis<i/>)
L70-72: Monopartite genomes consist of a circular, ssDNA molecule (similar to the DNA-A component of bipartite genomes) that is often associated with an alpha- or betasatellite, while bipartite...
L78-79: For instance, virions of bipartite begomoviruses encapsulate two circular, ssDNA components in which the two components are required for infectivity. But such a critical association between the two components is often... (there is only one genome, comprised of two components; the analogy made later in the manuscript between DNA components and eukaryotic chromosomes is essentially correct; so, just like humans have a genome comprised of 23 chromosomes, bipartite begomoviruses have a genome comprised of two components)
L197: Surely yeasts are not protozoan ! They are classified in the Kingdom Fungi.
L222: This is a dangerous assumption. Acronyms can be tricky to remember and I would strongly suggest that even the most seasoned geminivirologists will get at least 20% of them wrong. For example, tomato can be abbreviated as T or To, tobacco as T or Tob. Also, many species names have several words and acronyms can get extremely complex (ToLCCNV, ToLCVNV, ToLCTHV). So unless the search engine can account for (and suggest) alternative forms, this apparently useful feature can actually be counter-productive.
L238: It says Clustal W at the top of Figure 2.
L279: 21,298 iterons and TATA-boxes ?? Considering that each DNA component has 2-3 iterons and no more than 2 TATA-boxes, I would expect no more than 3,500 of these elements for 669 "genomes" (which I assume are components). The authors should re-check this script as it is obviously misidentifying the elements.
L322: I can't advise the authors strongly enough: **please** use an image of a geminate particle in the home page, instead of an ebola virus particle. For example, the one here <https://www.mdidea.net/support/glossary/virus/Geminivirus.jpg> seems to be free to use.
L350: monopartite
L448: This is indeed "unusually long". Another example of the script failing to identify an iteron correctly (see comment on L279). See Argüelo-Astorga et al., Virology (1994) and Argüelo-Astorga & Ruiz-Medrano, Arch Virol (2001) for proper iteron identification.
L450, 455: What is YOM ?? The correct acronym is CLCuB. Incidentally, this sequence is incorrectly named: Cotton leaf curl *virus betasatellite* does not make sense. It either one type of agent (virus) or the other (betasatellite). So the correct name is Cotton leaf curl betasatellite.
L453, 455: I have no idea what this acronym stands for. See comment on L222.

Experimental design

No comment.

Validity of the findings

No comment.

Additional comments

All my comments are in Section 1.

---

## Round 0.2 · accepted · Accept

I am satisfied with your response to the reviewer's suggestions and comments, Therefore I am happy to accept the manuscript for publication. Thank you very much